# Assessment of Infection Prevalence and Intensity of Disease-Causing Parasitic Protozoans *Perkinsus marinus* and *Haplosporidium nelsoni* in Georgia Oysters

**DOI:** 10.3390/microorganisms11071808

**Published:** 2023-07-14

**Authors:** Sarah Batchelor, J. Scott Harrison, Stephen E. Greiman, Laura M. Treible, John M. Carroll

**Affiliations:** 1Department of Biology, Georgia Southern University, Statesboro, GA 30460, USA; batchelor.sarah7@gmail.com (S.B.); sharrison@georgiasouthern.edu (J.S.H.); sgreiman@georgiasouthern.edu (S.E.G.); 2Department of Marine Biology, Savannah State University, Savannah, GA 31404, USA; treiblel@savannahstate.edu

**Keywords:** eastern oyster, *Crassostrea virginica*, Dermo, *Perkinsus marinus* MSX, *Haplosporidium nelsoni*, disease

## Abstract

Eastern oysters, *Crassostrea virginica*, are ecologically and economically important coastal species which provide a commercially valuable food product while also improving water quality through filtration, protecting shorelines, and providing habitat. The protozoan parasites *Perkinsus marinus* and *Haplosporidium nesloni* commonly infect oysters along the United States Atlantic and Gulf coasts and have been linked to poor oyster health and mass mortality events. In this study, wild oysters were collected from multiple reefs within four tidal creeks along the coast of Georgia to investigate *P. marinus* and *H. nelsoni* prevalence and intensity, their potential impact on oyster health, and identify possible drivers of the parasites. A second study occurred on four sites on Sapelo Island, Georgia, with continuous water quality monitoring data to further elucidate potential drivers. Oyster density and condition index, a proxy for health, were measured, and parasites were quantified using a TaqMan probe based quantitative real-time PCR within gill tissue. Real-time PCR showed that 86% of oysters tested were infected by one or both parasites in the coast-wide survey, and 93% of oysters from Sapelo Island were also infected by one or both parasites. Prevalence and infection intensity for both *P. marinus* and *H. nelsoni* varied across sites. Overall impacts on oysters were complex—intensity was not linked to oyster metrics in the coastwide study, but oyster condition was negatively correlated with *P. marinus* prevalence in the Sapelo Island study. Several relationships between both parasites and water quality parameters were identified, providing valuable information about potential drivers that should be investigated further.

## 1. Introduction

Eastern oysters, *Crassostrea virginica*, are commercially and ecologically important bivalves that provide many important ecosystem services along the US Atlantic and Gulf coasts [1,2,3]. Oyster reefs help protect shorelines [4], provide essential habitat for juvenile fishes and some adult finfish in estuarine communities [1], and improve water quality and facilitate nutrient cycling [5]. Oysters are commercially harvested throughout their range and the fishery has historically supported coastal communities [2]. Unfortunately, populations have been in decline due to multiple threats including overharvest, pollution, habitat loss and degradation, climate change, and disease [6,7]. The collapse of oyster populations has devastating consequences on coastal communities, and as a result, oysters are the subject of management and restoration.

Oysters are susceptible to a wide range of disease-causing species, including protozoa, bacteria, and viruses [8]. Many of these disease-causing organisms have adverse effects on individual oysters and their ecosystem services. Often related to oyster density and dependent on the species, these epizootic organisms can spread via contact with the water column and intermediate hosts [8,9]. Oyster parasites and pathogens are particularly concerning for struggling populations. Two protozoan parasites, *Perkinsus marinus* and *Haplosporidium nelsoni,* are prevalent along the Atlantic and Gulf coasts of the United States and have been identified as causative agents of potentially fatal oyster diseases [10,11,12], including in Georgia [13]. Both diseases have caused mass mortality events, contributing to declining oyster populations, and in regions where prevalence and intensity are high, have limited oyster population recovery [7].

The parasite *P. marinus* is the causative agent for the oyster disease Dermo, which can reduce oyster health and lead to mass mortality events, as has been observed elsewhere along the Atlantic coast [12]. The transmission of this parasite occurs through the water column when infective stages are released by both live and dead oysters and subsequently ingested by other oysters [14,15]. In several studies, high intensity infections reduce growth, tissue condition, reproductive capacity, and eventually, can lead to oyster mortality [14,16]. Water temperature and salinity are also important drivers of *P. marinus* [12,17,18], although other water quality parameters (i.e., pollution) can also impact the prevalence of this parasite [19]. Lower infections and rates of mortality in juvenile compared to adult oysters are usually observed, presumably because *P. marinus* is transmitted via the water column and larger oysters filter larger volumes of water [12]. This parasite was first observed in Georgia in the 1960s and may have caused morality events in the 1980s [13] and its prevalence remains high, particularly in the intertidal [20].

The parasite that causes multinucleated sphere unknown (MSX) disease in oysters is the protozoan *H. nelsoni*. This parasite is found from Maine to Florida and has caused several mass mortality events throughout this range, the first of which was documented in the Delaware Bay in 1957 [10,21]. MSX disease in oysters leads to degraded oyster tissues and reduced oyster health [22]. The prevalence and intensity of *H. nelsoni* is strongly influenced by environmental factors, including temperature and salinity. The parasite is most common when salinity exceeds 15 psu [10] and primarily occurs during the summer when water temperatures are highest [11]. The exact mode of transmission and complete life cycle of *H. nelsoni* is unknown, suggesting that oysters may be an accidental host or that an intermediate host may play a role in transmission [10,23]. This parasite was first detected in Georgia in the 1980s [13].

Given the negative impacts both parasites can have on eastern oysters, it is critical to more fully understand potential drivers. The unpredictable nature of these pathogens makes it hard to produce models that can explain the variance in oyster abundances from year to year [24]. Rapid proliferation of both MSX and Dermo is correlated with temperatures in excess of 20 °C, as well as high salinities [8], which are already commonly experienced by oysters throughout the southeastern US and Gulf of Mexico coast. Climate change associated stressors will also likely increase the prevalence, intensity, and negative impacts of these pathogens [25]. Both diseases are now common in the coastal waters of Georgia, USA, where water temperatures exceed 20 °C for over half the year. *Perkinsus marinus* was first isolated from oyster tissues at low prevalence in the 1960s, while *H. nelsoni* was first isolated from oyster tissues in the 1980s [13]. Both parasites have been commonly found in high prevalence more recently [20,26,27], although their current impacts are unclear. *Perkinsus marinus* is considered the causative agent of mass mortality events in the late 1980s throughout Georgia [13], although the disease has been found at high prevalence since (>75% of oysters), with seemingly limited impacts on oyster local health [20,26].

For oyster restoration and management, it is essential to understand the prevalence and intensity of pathogen causing protozoans in Georgia and whether they negatively impact local oyster populations. Therefore, the overarching goal for this study was to examine the potential impacts of protozoan parasites on oyster populations along the Georgia coast. We had three main objectives; (1) to determine the modern prevalence and intensity of both protozoan parasites along the Georgia coast, (2) to examine oyster health across sites along the coast and explore whether infection type prevalence or intensity affect oyster condition, and 3) to explore potential environmental drivers of both parasites.

## 2. Materials and Methods

### 2.1. Coastwide Sampling

In 2018, oysters were harvested from between 8 and 11 reefs within each of the four tidal creeks along the Georgia coast: Oyster Creek (32°0.441′ N 80°54.722′ W; 25 July), Medway River (31°43.775′ N 81°13.246′ W; 20 July), Teakettle Creek (31°27.334′ N 81°18.340′ W; 24 July), and Jointer Creek (31°5.219′ N 81°29.209′ W; 7 August; Figure 1) in late July to early August. At each reef location within the creek, water quality measurements (T, S, DO, and pH) were measured using a handheld YSI ProDSS sonde and averaged for each creek. Although these represent just a single snapshot of water quality, the creeks sampled exhibit fairly consistent gradients in water quality features due to physical and geomorphic characteristics of the creeks [28]. Three 0.25 × 0.25 m quadrats were harvested for oysters from each reef and frozen prior to processing. At the lab, oyster individuals were separated and cleaned, then measured for shell length and width, weighed, and dissected. A small gill tissue sample was excised, weighed, preserved in 95% ethanol and stored at −20 °C until later extraction. The remaining tissues and shell were dried in an oven at 70 °C for ~48 h and reweighed to determine condition index. Condition indices for each oyster were calculated using the formula:$$Condition\;index=\frac{DTW}{WW-DSW}\times 100$$
where *DTW* is the dry tissue weight, *WW* is the whole wet weight of the oyster and shell, and *DSW* is the dry sell weight. This metric of fatness is used as a proxy for oyster health.

Genomic DNA was extracted from gill tissue using a Zymo Quick-DNA Plant/Seed Miniprep Kit (Zymo Research, Irvine, CA, USA) following the manufacturer’s recommended protocol. These kits were selected because they effectively removed polysaccharide PCR inhibitors found within oyster tissues [29]. DNA extractions were quantified using a Quibit^TM^ dsDNA High Sensitivity Assay kit (ThermoFisher Scientific, Waltham, MA, USA) and if possible, DNA extractions were standardized to 50 ng/µL.

Infection prevalence and intensity of *P. marinus* and *H. nelsoni* were determined using TaqMan probe based Quantitative real-time PCR (qPCR) assays. Quantitative real-time PCR reactions were run on an ABI StepOne Plus real time PCR machine using the primers, TaqMan probes, and protocols for *P. marinus* (ITS1 gene; [30]) and for *H. nelsoni* (18S rDNA gene; ref. [31]). To generate the standard curves necessary for quantification of parasite intensities, laboratory synthesized double stranded DNA fragments (GeneStrands) overlapping the amplicons for *P. marinus* and *H. nelsoni* were used (Eurofins Genomics, Louisville, KY, USA). The gene fragments were delivered dry, and the total copy number for each gene was calculated using the formula: number of copies = (dry weight of DNA fragment (ng) × 6.022 × 10^23^)/(length of fragment × 1 × 10^9^ × 650). This formula assumes that the average weight of a base pair is 650 Daltons. The dried gene fragments were suspended in sterile TE buffer to a stock concentration of 1.0 × 10^10^ copies/µL. From the stocks, 7 10-fold dilutions were made ranging from 10 to 10,000,000 gene copies and used to construct standard curves. DNA standards and negative controls (DNA free water) were run in triplicate DNA. Critical threshold (Ct) and gene copy number values for samples were calculated by the StepOnePlus V2.3 software. Results from qPCR indicate the quantity of parasite cell copies per 200 ng of total genomic DNA.

### 2.2. Sapelo Island Sampling

The 2018 samples displayed high but variable prevalence and intensity of both parasites (see results), but we only had a single time point for water quality in these creeks. To supplement the initial survey and explore relationships between diseases and water quality, oysters were sampled at four sites around Sapelo Island on 1 September 2020. We selected sites to coincide with the Sapelo Island National Estuarine Research Reserve (SINERR) System-Wide Monitoring Program (SWMP) stations that continuously collect water quality data (T, S, DO, pH) every 15 min. The sites at SINERR were Hunt Camp, Cabretta Creek, Dean Creek, and Ferry Dock (Figure 1 inset). Three quadrats of oysters were harvested from each location and processed for condition and parasite detection as described above. Water quality data for the three months prior to collection was accessed from the NOAA National Estuarine Research Reserve System (NERRS) Central Data Management Office on 23 February 2023. We selected the 3-month period prior to the 1 September collection date since both parasites are most abundant in the summer and are typically related to water quality conditions found during the summer (i.e., warm temperatures).

### 2.3. Data Analyses

To determine impacts of environmental conditions on parasite prevalence in creeks sampled in 2018, parasite prevalences (Dermo and MSX) were fitted with logistic mixed models using the ‘glmer’ function in R (estimated using ML and BOBYQA optimizer). Dermo prevalence (yes/no) and MSX prevalence (yes/no) were modeled as a function of temperature, salinity, pH, and D.O. as fixed effects. The models included site and creek as random effects with sites nested within each creek to account for psuedoreplication within creeks.

To determine whether infection varied with environmental conditions across creeks, we used only oysters with parasites present. Linear mixed models were used to model parasite intensity using the ‘lmer’ function in R (estimated using REML and BOBYQA optimizer). Intensity values (copies per 200 ng of tissue) for both Dermo and MSX were modeled as a function of temperature, salinity, pH, and DO as fixed effects. All models included site and creek as random effects with sites nested within each creek to account for psuedoreplication within creeks.

To determine whether parasite infection impacted oysters, oyster condition index and oyster length were modeled as a function of parasite prevalence of both Dermo and MSX as fixed effects. Creek was also included as a fixed effect to determine potential differences in oysters within different creeks, and site was included as a random effect. For all linear models, Normality of residuals was confirmed using quantile–quantile plots [32], and homoscedasticity was confirmed with standardized residual plots.

Analysis for 2020 samples around Sapelo Island generally followed the same framework described above. However, rather than point measurements for water quality, we used continuously monitored water quality data from the Sapelo Island NERR SWMP stations. For these stations, we calculated the mean temperature, salinity, DO, pH, and turbidity, logged every 15 min, from the three months prior to the collection of oysters on 1 September 2020. Parasite prevalence models were fitted with logistic mixed models as a function of temperature, salinity, and DO as fixed effects and creek as a random effect. Linear models were also used to predict oyster condition and oyster length as a function of parasite prevalence and creek.

## 3. Results

### 3.1. Coastwide Survey

In 2018, we sampled 32 oysters in Jointer Creek, 31 each from Oyster and Teakettle Creeks, and 28 oysters from the Medway River. Across all creeks in 2018, 86% of oysters sampled were infected with one or both disease-causing parasites, and 53% were co-infected. Including co-infected oysters, 61% of oysters were infected by *P. marinus* (Dermo) and 77% of oysters were infected by *H. nelsoni* (MSX). For Dermo, infection rates were 68%, 39%, 81%, and 56% for Oyster, Medway, Teakettle, and Jointer Creeks, respectively. Infection rates for MSX were 68%, 64%, 87%, and 88% for Oyster, Medway, Teakettle, and Jointer Creeks, respectively. Across all sites, when *P. marinus* was absent, *H. nelsoni* infected 64% of oysters, compared to 85% of oysters also hosting *P. marinus*. Likewise, *P. marinus* only infected 39% of sampled oysters when *H. nelsoni* was absent, but 68% of oysters when *H. nelsoni* was present. Chi-squared test reveals a higher-than-expected probability of coinfection between the two parasites (Χ^2^ = 7.55, *p* < 0.01).

Despite overall high prevalence across the creeks, parasite intensity was highly variable. Across all creeks, when present, Dermo detection ranged from ~1 to over 1 million parasite gene copies per 200 ng of tissue. Similarly, for MSX, detection ranged from ~1 to over 12.4 million gene copies per 200 ng of tissue. Intensity was similarly variable within and among creeks. When Dermo was detected, infection intensity was 752 ± 1903 in Oyster Creek, 29.7 ± 9.5 in Medway, 44,420 ± 201,555 in Teakettle Creek, and 9880 ± 37,341 in Jointer Creek. Similarly, when MSX was detected, infection intensity was 199 ± 407 in Oyster Creek, 943 ± 3198 in Medway, 133,605 ± 485,522 in Teakettle Creek, and 1,000,348 ± 2,932,110 in Jointer Creek. When controlling for creek, there were no impacts of any of the tested environmental variables (temp., salinity, D.O., or pH) on prevalence of either Dermo or MSX (all *p* > 0.05, Table 1 and Table 2). Similarly, MSX intensity was not impacted by any of the tested environmental variables (all *p* > 0.05, Table 3), but Dermo intensity was significantly predicted by salinity (*p* < 0.001), D.O. (*p* = 0.003), and pH (0.002) (Table 4).

Oyster condition index, a metric for oyster health, varied across creeks (Figure 2). When comparing across all creeks, there was a trend with oysters with neither parasite having the highest condition, but there was no significant effect of Dermo presence (*p* = 0.928) or MSX presence (*p* = 0.074) on oyster condition (Table 5). However, there was an effect of creek, with Oyster Creek samples having significantly lower condition (*p* = 0.006), potentially due to the highest salinities at this site. Oyster size (length, mm) was not significantly related to either Dermo presence (*p* = 0.142) or MSX presence (*p* = 0.842; Table 6).

### 3.2. Sapelo Island Survey

In 2020, we sampled 29 oysters from Cabretta Creek, 26 oysters from Dean Creek, 25 oysters from Ferry Dock, and 31 oysters from Hunt Camp. Across all creeks, 93% of oysters sampled were infected with one or both disease-causing parasites, and 51% were co-infected. Including co-infected oysters, 82% of oysters were infected by Dermo and 60% of oysters were infected by MSX. For Dermo, infection rates varied among creeks with 59%, 81%, 92%, and 97% of oysters infected in Cabretta, Dean Creek, Ferry Dock, and Hunt Camp, respectively. Likewise, for MSX, infection rates were 62%, 85%, 92%, and 13% for Cabretta, Dean Creek, Ferry Dock, and Hunt Camp, respectively. Unlike the coastwide survey, the observed number of coinfected individuals was not different than the expected number (Χ^2^ = 0.29, *p* = 0.59). When controlling for creek, there were no impacts of any of the tested environmental variables (temp., salinity, or D.O.) on prevalence of Dermo (all *p* > 0.05, Table 7). However, MSX prevalence was significantly predicted by temperature (*p* < 0.001), salinity (*p* = 0.049), and D.O. (*p* = 0.050, Table 8).

Intensity of Dermo was relatively high across all Sapelo sites in 2020. When Dermo was detected, infection intensity was 1018 ± 2556 in Cabretta Creek, 823 ±1340 in Dean Creek, 13,196 ± 42,400 in Ferry Dock, and 3880 ± 15,234 in Hunt Camp. Similarly, when MSX was detected, 1591 ± 6567 in Cabretta Creek, 49,626 ± 23,271 in Dean Creek, 1962 ± 6052 in Ferry Dock, and 5.6 ± 6.4 in Hunt Camp.

Oyster condition was significantly impacted by site, where the oysters found at Cabretta Creek (*p* < 0.001), which was the saltiest site with the highest DO, had higher condition than oysters at other sites. Additionally, the condition was highest in oysters without either parasite at Cabretta Creek (*p* < 0.001, Table 9). Oyster length (mm) was impacted by creek, with Dean Creek (*p* = 0.005) and Ferry Creek (*p* = 0.003) having the shortest oysters. Additionally, oysters infected with Dermo were larger than oysters not infected (*p* = 0.017, Figure 3, Table 10).

## 4. Discussion

The protozoan parasites *P. marinus* and *H. nelsoni* are prevalent along the US Atlantic and Gulf coasts and, when infections are severe, are the causative agents of potentially fatal oyster diseases Dermo and MSX, respectively. Our study indicates that both parasites are common in coastal Georgia, with over 85% of all oysters sampled hosting at least one of these disease-causing parasites. Despite the high overall prevalence, there was some among-site variability in the presence of each parasite ranging from as low as 12% to as high as 97% prevalence. When the parasites were detected, infection intensity also varied within and among sites, with some individuals having parasite loads in the millions. Despite the high abundance in oysters, there appears to be limited effect of the parasites on oyster condition index suggesting that while very common, these two protozoans may not lead to disease in intertidal oysters in coastal Georgia.

The high prevalence and intensity of the parasites in coastal Georgia may be expected—samples were collected during the summer when temperatures peak and parasite loads are expected to be at their highest. High overall prevalence and variable intensity is common for both parasites in southeastern estuaries, including in North Carolina [31], South Carolina [33], Georgia [13,20], and Florida [34]. In Georgia, *P. marinus* was first detected in 1966, with anywhere from 0–44% of oysters infected depending on the site [13]. During follow-up surveys in 1986–1987, Lewis et al. (1992) found that *P. marinus* had become highly prevalent, occurring in 88–100% of their sampled populations, and detected *H. nelsoni* for the first time in 2–6% of the sampled population. However, in surveys since, *H. nelsoni* has also increased in prevalence in Georgia [20,27].

There are several reasons explaining the overall high prevalence of both parasites in Georgia. Both parasites thrive in warm (above 20 °C) and meso- to euryhaline (above 15 psu) waters [21,35], which occurred at all study sites in both survey years. Additionally, oysters in Georgia are almost exclusively intertidal, and tidal location can also influence disease prevalence and the risk of mortality events [8]. High parasite prevalence and intensity with associated disease probability are also linked to intertidal air exposure [20], where most oysters in Georgia and throughout the southeast live [36]. Exposure to air can negatively impact an oyster’s immune response to pathogens, increasing their susceptibility to disease [37,38,39]. Exploring across field sites from Maine to North Carolina, researchers determined that *P. marinus* prevalence was higher in intertidal oysters when compared to subtidal oysters [40]. Similarly, while there was no effect of intertidal location on *P. marinus* prevalence in Georgia, intensity of *P. marinus* and prevalence of *H. nelsoni* were higher in the intertidal locations [20]. However, not all studies of parasite prevalence in oysters find the same patterns between intertidal and subtidal oysters [41], and we only sampled intertidal oysters in this study.

Despite a generally high prevalence across all sites in both years, intensity of infection by both parasites was variable within and among sites. While only a few oysters had no detectable parasite loads, there was a range of just a few parasite gene copies detected in an individual oyster to over 10 million parasite gene copies detected. While it is important to note that gene copies are not the same as individual parasites, both parasites are unicellular organisms, so gene copies are a good proxy for relative parasite density. Variation in presence and intensity of *P. marinus* has been attributed to oyster length in some studies where infections are less intense in smaller juvenile oysters compared to adult oysters [12]. Since the primary mechanism for infection of *P. marinus* is via filtration, larger and older oysters can filter more water and have been exposed for longer time periods, which could lead to greater exposure risk with size [12]. At the Sapleo sites in 2020, we found a significant relationship between Dermo presence and oyster size (length), with higher disease prevalence in longer oysters. However, we did not find similar patterns with oyster size in the coastwide study in 2018. *Haplosporidium nelsoni* has been demonstrated to infect both juvenile and adult oysters alike [23], and our study confirms that there appears to be no relationship between oyster size and the prevalence or intensity of *H. nelsoni*.

The prevalence and intensity of these disease-causing parasites appears to be site-specific and may be partly driven by environmental conditions. In both the coastwide and Sapelo Island studies, the prevalence of *P. marinus* was not significantly affected by any of the water quality variables tested. The lack of environmental relationships were unexpected, especially for *P. marinus,* which typically increases with increasing temperature, and can proliferate in temperatures up to 35 °C [14]. However, temperature was generally high across all sites in both years and the limited variability in creek temperature could mask any relationships. When we conducted the Sapelo Island surveys and were able to leverage longer, continuously monitored water quality data, there was a negative relationship between salinity and *P. marinus* prevalence that trended toward significance (*p* = 0.06). Unlike the prevalence surveys, *P. marinus* intensity showed a negative response to salinity and dissolved oxygen when accounting for other water quality variables, in the coastwide survey. While other studies have found that Dermo may increase with salinity [34,42], it is possible that Dermo may peak at intermediate–high salinities and decrease as salinity becomes oceanic [43]. Likewise, lower oxygen conditions can affect oyster immune responses, leading to higher prevalence, intensity, and progression of Dermo disease in oysters [17,39].

The relationships between *H. nelsoni* and water quality parameters were complex and varied depending on the survey. In the coastwide survey, there was no impact of spot water quality measurements *on H. nelsoni* prevalence or intensity. Temperature and salinity are both also linked to MSX prevalence and intensity [23,44], so it is possible that there was not enough variation in these water quality variables among sites and creeks to observe a pattern. For the Sapelo surveys, we found that temperature, salinity, and DO all affected the prevalence of *H. nelsoni*. Interestingly, when controlling for other variables, the prevalence of *H. nelsoni* increased with decreasing temperature and salinity while increasing with increasing DO. The temperature was relatively similar across all sites in the study period (i.e., mean T 29.2–29.9 °C), so any temperature effect is likely confounded by other site characteristics. Although we might expect *H. nelsoni* to increase with salinity [45], the salinity across the sites was also relatively high—from ~24 at Hunt Camp to ~31 at Cabretta Creek—and MSX might be highest at intermediate salinity [44]. Further, the sites with the highest MSX also experienced the greatest variance in salinity over the preceding three-month period, suggesting these infections may be more common in oysters that experience greater variability in water quality conditions, which could increase stress for oysters and potentially increase their susceptibility to disease-causing parasites [46].

It is important to note that a large proportion of oysters in this study were coinfected by both parasites—52% in the coastwide survey and 50% on Sapelo Island. Since the geographic range of both parasites overlaps, and both parasites have similar relationships with water quality variables, coinfection is expected to be common [20,47], as with other oyster parasites [34,48]. Coinfection typically occurs in one of two ways, either hosts are independently infected by multiple parasites at the same time, or infections can happen sequentially [49]. Often, the presence of one parasite may alter the host’s immune system and potentially increase susceptibility to additional parasites and pathogens [47,48,50]. It is unclear with which pathway coinfection is occurring in Georgia. In the coastwide survey, oysters with one parasite were significantly more likely to have the other, suggesting a possible sequential coinfection and increased vulnerability. Since evidence suggests that small juvenile oysters may be infected by *H. nelsoni* [23], whereas *P. marinus* is more common in larger, older oysters [12], it is possible that *H. nelsoni* may facilitate the infection of oysters by *P. marinus*. However, in the Sapelo survey, the observed coinfection was not higher than expected. Overall infection by *P. marinus* was much higher in the Sapelo samples collected in 2020 (82%) than in the coastwide samples collected in 2018 (61%), but coinfection was similar across both surveys, so it is possible that spatiotemporal variability in either parasite could impact coinfection patterns. The mechanism for coinfection, the relationships between coinfection and environmental parameters, and the impacts on oysters should be investigated further.

Interestingly, oyster condition, the metric commonly used as a proxy of oyster health, appears minimally impacted by the presence and intensity of these disease-causing parasites. In the coastwide sampling, there was no relationship between the prevalence or intensity of either parasite or oyster condition. Instead, site water quality characteristics appear to be stronger drivers of overall oyster health—oyster condition index decreased as the salinity increased. This is not surprising, as oyster performance tends to be optimized in intermediate salinity [12,51,52], including in Georgia [36]. This may be impacted by differences between creeks, as overall, Oyster Creek oysters exhibited the lowest condition, but this was also the creek with the highest salinity. In the Sapelo Island sampling, the prevalence or intensity of either parasite did not negatively impact oyster condition. However, there was an effect of creek on oyster condition, with Cabretta Creek oysters exhibiting the highest condition. In contrast to the coastwide sampling, Cabretta Creek oysters experience the highest salinity; however, they also experience the coolest temperatures and, generally, the most stable conditions. We also did not observe any potential negative impacts of coinfection on oyster condition, i.e., there were no apparent additive or synergistic effects from either parasite. Thus, our results highlight that potentially complex and confounding interactions between water quality variables are more important than parasite prevalence or intensity for oyster health, at least in Georgia, and should be explored further.

The lack of negative impacts *P. marinus* and *H. nelsoni* have on oyster health found in this study are inconsistent with prior studies. Oysters are susceptible to a wide range of pathogen-causing species, including microscopic organisms like protozoa, bacteria, and viral infections [8], which can have adverse effects on oysters and their ecosystem services [12,22,53]. Although low-level infections can be non-lethal [54], negative relationships between both parasites and oyster condition have been observed repeatedly [16,53,55]. Additionally, these parasitic infections have been attributed to several oyster mass mortality events [11,56,57], including historically in Georgia [13]. This study suggests that the prevalence of *P. marinus* and *H. nelsoni* on the Georgia coast appear to have increased since their first occurrences in Georgia oyster populations [13]. Despite this apparent increase, subsequent studies in coastal Georgia since that period have similarly not detected reduced health or mortality in disease-infected oysters [20,27,41,43], including the current study. Thus, it is possible that local oyster populations in Georgia are resistant or more resilient to infections from these parasites [58,59] or tolerant of co-stressors (i.e., temperature) that might lead to reduced oyster health and even death.

In this study, we used modern molecular techniques to explore parasite prevalence and intensity, whereas historically, the primary method of investigating infection intensity was histology. However, histology is dependent upon the life stage of the parasites and an oyster response to be detected [30]. Using qPCR allows for more a sensitive and accurate detection of parasite presence while also quantifying intensity whether or not parasites are at an appropriate and infective life stage [30,31]. Therefore, it is possible that the prevalence and intensity of these parasites has not changed over time, but detection sensitivity has increased. Low intensity infections detected in this study may be cases where parasites have not fully developed, since qPCR uses DNA which is present throughout the parasite life cycle [31] and, therefore, may not be impacting health or condition at the point in time when the oysters were collected. Further, it is possible that high prevalence may be common, but these infections may not be symptomatic, reducing the probability of disease detection using traditional methods. Regardless, the occurrence of both disease-causing parasites appears ubiquitous along the Georgia coastline, as well as elsewhere in the southeast US [20,31,34].

The ecological and economic consequences of disease-causing parasites may be significant, so the interplay of water quality and parasites on oyster health, as well as the drivers of parasite prevalence and intensity, must continue to be explored. While both studies presented here represent snapshots of parasites and water quality, the results suggest that the relationships are quite complex and may vary across spatial scales observed and across years, highlighting the need for long-term data [60]. Water quality can impact oyster health and disease susceptibility [46], as well as the life cycles of the parasites themselves [14,23,61], and may interact with several additional variables not explored in this study, like location within a seascape. Within the context of climate change, it becomes increasingly important to explore sensitivity of disease-causing parasites to temperature, salinity, oxygen, and pH to better predict how climate-induced changes will influence these host–parasite interactions [62]. For example, high winter temperatures can affect parasite prevalence and intensity in the subsequent summer [35], and increasing environmental variability can influence host physiology and immune responses [63], and both stressors are expected to increase with climate change. Future studies should explore variables with higher spatiotemporal resolution across multiple years, incorporate immune response and gene expression, and use a stepwise, hierarchical approach to analysis to fully elucidate the complexities associated with disease-causing parasites and their interactions with oyster hosts.

## Figures and Tables

**Figure 1 microorganisms-11-01808-f001:**
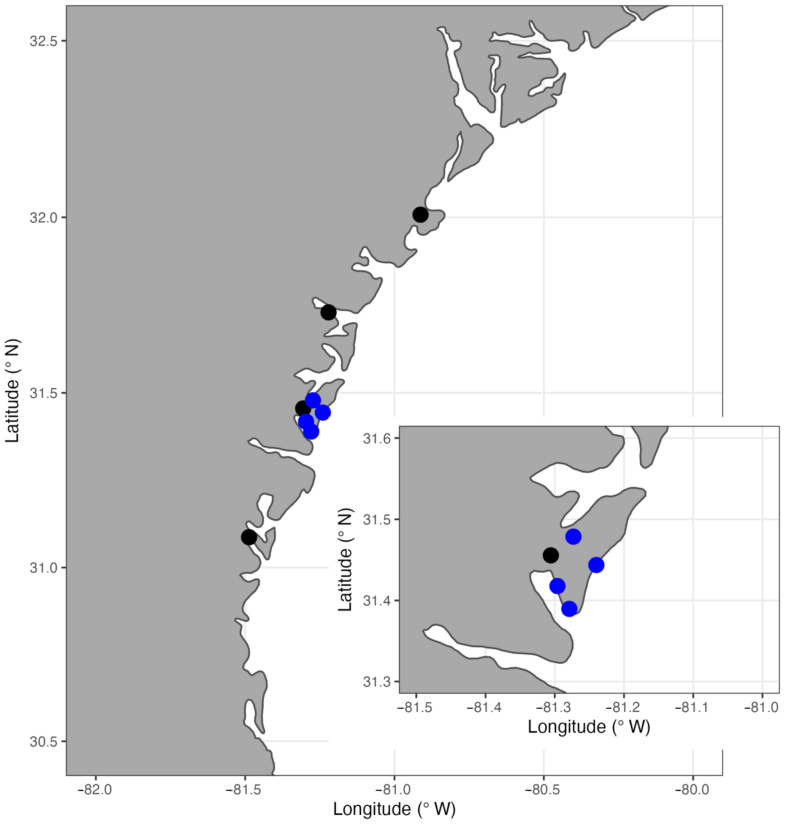
Map of the Georgian coastline showing the 2018 collection sites (black circles) for the coastwide sampling. Inset shows a closer view of Sapelo Island and the four System Wide Monitoring Program stations (blue circles) within the Sapelo Island National Estuarine Research Reserve where oysters were collected in 2020.

**Figure 2 microorganisms-11-01808-f002:**
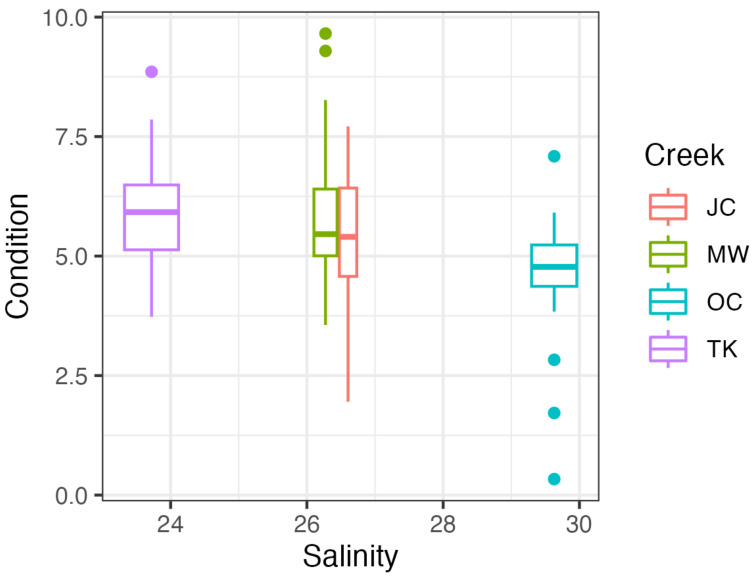
Condition index of oysters collected in the Georgian coastwide survey across the four tidal creeks across the salinity gradient.

**Figure 3 microorganisms-11-01808-f003:**
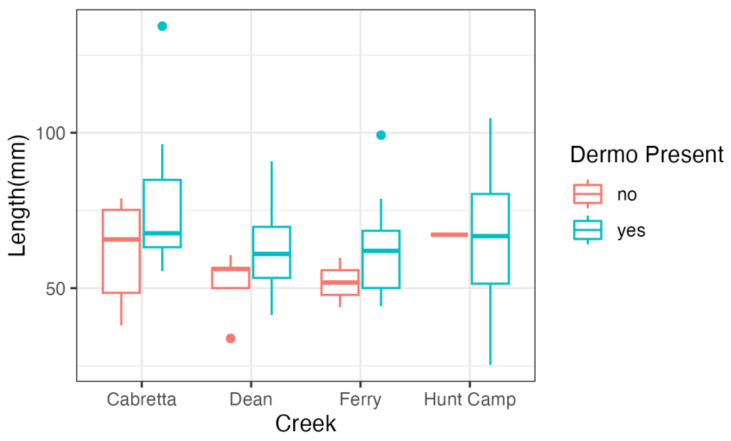
Relationship between oyster size (mm) and presence or absence of *Perkensis marinus* (Dermo) across the four study creeks in the Sapelo Island survey. Oysters with parasites are larger.

**Table 1 microorganisms-11-01808-t001:** Logistic mixed model results predicting Dermo prevalence as a function of environmental variables for the coastwide survey (2018). D.O.: dissolved oxygen.

Fixed Effect	Estimate	Std. Error	z Value	*p*-Value
(Intercept)	−44.77	28.839	−1.552	0.121
Temperature	−0.291	0.204	−1.423	0.155
Salinity	−0.258	0.162	−1.592	0.111
D.O.	−0.263	0.611	−0.431	0.666
pH	8.392	4.683	1.792	0.073

**Table 2 microorganisms-11-01808-t002:** Logistic mixed model results predicting MSX prevalence as a function of environmental variables for the coastwide survey (2018). D.O.: dissolved oxygen.

Fixed Effect	Estimate	Std. Error	z Value	*p*-Value
(Intercept)	5.4556	36.666	0.149	0.882
Temperature	0.030	0.266	0.114	0.910
Salinity	−0.249	0.184	−1.353	0.176
D.O.	0.277	0.801	0.345	0.730
pH	0.065	5.754	0.011	0.991

**Table 3 microorganisms-11-01808-t003:** Linear mixed model results predicting Dermo intensity as a function of environmental variables for the coastwide survey (2018). Values in italics are significant at *p* < 0.05. D.O.: dissolved oxygen.

Fixed Effect	Estimate	Std. Error	t Value	*p*-Value
(Intercept)	−2.73 × 10^6^	1,090,546	−2.504	*0.014*
Temperature	−8964.32	9125	−0.982	0.328
Salinity	−19,756.61	5704	−3.464	<*0.001*
D.O.	−68,001.38	22,109	−3.076	*0.003*
pH	5.19 × 10^5^	166,146	3.123	*0.002*

**Table 4 microorganisms-11-01808-t004:** Linear mixed model results predicting MSX intensity as a function of environmental variables for the coastwide survey (2018). D.O.: dissolved oxygen.

Fixed Effect	Estimate	Std. Error	t Value	*p*-Value
(Intercept)	−5.13 × 10^6^	19,660,965	−0.261	0.795
Temperature	−19,362.58	249,893	−0.077	0.938
Salinity	−24,539.51	163,048	−0.151	0.881
D.O.	−44,266.66	382,837	−0.116	0.908
pH	9.20 × 10^5^	2,798,761	0.329	0.743

**Table 5 microorganisms-11-01808-t005:** Linear mixed model results predicting oyster condition as a function of disease presence and creek for the coastwide survey (2018). Values in italics are significant at *p* < 0.05. MW: Medway Creek, OC: Oyster Creek, TK: Teakettle Creek.

Fixed Effect	Estimate	Std. Error	t Value	*p*-Value
(Intercept)	5.928	0.359	16.496	<*0.001*
Dermo present	0.024	0.265	0.090	0.928
MSX present	−0.554	0.307	−1.806	0.074
Creek-MW	0.192	0.348	0.552	0.582
Creek-OC	−0.944	0.339	−2.783	*0.006*
Creek-TK	0.439	0.339	1.293	0.199

**Table 6 microorganisms-11-01808-t006:** Linear mixed model results predicting oyster length (mm) as a function of disease presence and creek for the coastwide survey (2018). Values in italics are significant at *p* < 0.05. MW: Medway Creek, OC: Oyster Creek, TK: Teakettle Creek.

Fixed Effect	Estimate	Std. Error	t Value	*p*-Value
(Intercept)	62.373	6.607	9.441	<*0.001*
Dermo present	6.892	4.665	1.477	0.142
MSX present	−1.097	5.476	−0.200	0.842
Creek-MW	17.752	6.368	2.788	*0.006*
Creek-OC	6.089	6.246	0.975	0.332
Creek-TK	31.845	6.142	5.185	<*0.001*

**Table 7 microorganisms-11-01808-t007:** Logistic mixed model results predicting Dermo prevalence as a function of environmental variables for the Sapelo Island survey (2020). D.O.: dissolved oxygen.

Fixed Effect	Estimate	Std. Error	z Value	*p*-Value
(Intercept)	0.493	59.617	0.008	0.993
Temperature	0.452	1.954	0.231	0.817
Salinity	−0.659	0.355	−1.855	0.064
D.O.	1.312	1.360	0.965	0.335

**Table 8 microorganisms-11-01808-t008:** Logistic mixed model results predicting MSX prevalence as a function of environmental variables for the Sapelo Island survey (2020). Values in italics are significant at *p* < 0.05. D.O.: dissolved oxygen.

Fixed Effect	Estimate	Std. Error	z Value	*p*-Value
(Intercept)	228.183	50.377	4.530	<*0.001*
Temperature	−7.473	1.596	−4.682	<*0.001*
Salinity	−0.721	0.366	−1.971	*0.049*
D.O.	2.729	1.391	1.962	*0.050*

**Table 9 microorganisms-11-01808-t009:** Linear model results predicting oyster condition as a function of disease presence and creek for the Sapelo Island survey (2020). Values in italics are significant at *p* < 0.05.

Fixed Effect	Estimate	Std. Error	t Value	*p*-Value
(Intercept)	7.657	0.935	8.183	<*0.001*
Dermo present	−0.277	0.923	−0.299	0.765
MSX present	−0.647	0.873	−0.741	0.460
Creek-Dean	1.826	0.949	1.924	0.057
Creek-Ferry	−1.021	0.993	−1.028	0.306
Creek-Hunt Camp	−1.810	1.062	−1.705	0.091

**Table 10 microorganisms-11-01808-t010:** Linear model results predicting oyster length (mm) as a function of disease presence and creek for the Sapelo Island survey (2020). Values in italics are significant at *p* < 0.05.

Fixed Effect	Estimate	Std. Error	t Value	*p*-Value
(Intercept)	61.237	4.384	13.970	<*0.001*
Dermo present	10.512	4.326	2.430	*0.017*
MSX present	3.889	4.089	0.951	0.344
Creek-Dean	−12.887	4.447	−2.898	*0.005*
Creek-Ferry	−14.068	4.654	−3.023	*0.003*
Creek-Hunt Camp	−5.656	4.973	−1.137	0.258

## Data Availability

Data available upon request.

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
