# Peer review of "Assessment of Infection Prevalence and Intensity of Disease-Causing Parasitic Protozoans Perkinsus marinus and Haplosporidium nelsoni in Georgia Oysters"

_microorganisms, 2023, doi:10.3390/microorganisms11071808_

Round 1

Reviewer 1 Report

Demo and MSX are among the most significant mollusk pathogens for oyster aquaculture industry in the USA, which have been also concerned by many mollusk researchers around the world. The authors here investigated the prevalence and intensity, and their potential impact on Eastern oysters along the coast of Georgia. The study was well designed, and the methods employed for analysis is qualified. The main results has been presented properly. While there are still several points need to be revised before publication.

1. Some important information is missed in the abstract. In 2018 coastwide survey of multiple sites, there is no relation between parasite infection. While in 2020 Sapelo Island survey, oyster conditions have a strong, negative correlation with Dermo prevalence. The 2020 survey results should also be present in the abstract. And do the authors have any hypothesis to explain the inconsistencies, e.g. annual changes, site variations? This should be important and interesting for the understanding of the factors related to disease occurrence and oyster healthy.

2. Introduction

The authors have summarized the main research progress associated with the two diseases. While the paragraphs were not well organized. The authors should pay more attention to the ogical consistency. It seems that the 3rd and 4th paragraphs are isolated from the other parts of introduction.

The meaning of Lines 86-90 should be rechecked, there are many contradictions. I am confused whether there are negative impacts of the two infectious diseases to oysters health?

Some words should be missed after ‘since’ in line 90.

3. Results

Please give separate titles of Y axis of A and B plates in Figure 2, as showed in Figure 4. It will be better if the author could denote whether there are significant difference between different bars in each plates of figure 2 and 4.

4. Discussion

As I have mentioned previously, the author did not provide sufficient attention to the different results in 2020 Sapelo Island survey compared to the 2018 coastwide survey. In the discussion section, the author should discuss this point specifically.

1. what is the specific meaning of "gulf coast" in lines 42 and 81.

2. the genus name should be abbreviated after their first appearence.

Author Response

Reviewer 1

Comments and Suggestions for Authors

Demo and MSX are among the most significant mollusk pathogens for oyster aquaculture industry in the USA, which have been also concerned by many mollusk researchers around the world. The authors here investigated the prevalence and intensity, and their potential impact on Eastern oysters along the coast of Georgia. The study was well designed, and the methods employed for analysis is qualified. The main results has been presented properly. While there are still several points need to be revised before publication.

We appreciate the Reviewer feedback and we feel we have addressed the issues highlighted below in this new version.

  1. Some important information is missed in the abstract. In 2018 coastwide survey of multiple sites, there is no relation between parasite infection. While in 2020 Sapelo Island survey, oyster conditions have a strong, negative correlation with Dermo prevalence. The 2020 survey results should also be present in the abstract. And do the authors have any hypothesis to explain the inconsistencies, e.g. annual changes, site variations? This should be important and interesting for the understanding of the factors related to disease occurrence and oyster healthy.

We apologize for omitting Sapelo from the abstract and have now fixed this issue. We currently don’t have a good explanation for inconsistencies between coastwide and Sapelo results.  Disease dynamics are complex, and a suite of other factors might also influence the overall effect of parasites on hosts. We agree that it could be due to both site effects and annual variation and hope we have improved the discussion regarding these discrepancies.

  1. Introduction

The authors have summarized the main research progress associated with the two diseases. While the paragraphs were not well organized. The authors should pay more attention to the ogical consistency. It seems that the 3rd and 4th paragraphs are isolated from the other parts of introduction.

The meaning of Lines 86-90 should be rechecked, there are many contradictions. I am confused whether there are negative impacts of the two infectious diseases to oysters health?

Some words should be missed after ‘since’ in line 90.

We have made major changes to the introduction based on both this Review and comments from the other Reviewer.  We have removed a paragraph from the introduction, reworked paragraphs 3 and 4 to make the text flow better and clarified confusing statements.

  1. Results

Please give separate titles of Y axis of A and B plates in Figure 2, as showed in Figure 4. It will be better if the author could denote whether there are significant difference between different bars in each plates of figure 2 and 4.

 We have reorganized the manuscript and the figures.

  1. Discussion

As I have mentioned previously, the author did not provide sufficient attention to the different results in 2020 Sapelo Island survey compared to the 2018 coastwide survey. In the discussion section, the author should discuss this point specifically.

We have made major changes to the discussion due to comments from both Reviewers.  Importantly, based on the suggestions from Reviewer 2, we used a different analytical approach which removed any relationship between Dermo and condition in the Sapelo surveys.  However, we have made sure to include when and where things were different between the surveys throughout the discussion.

Comments on the Quality of English Language

  1. what is the specific meaning of "gulf coast" in lines 42 and 81.

The US Gulf coast.

  1. the genus name should be abbreviated after their first appearence.

We have fixed this.

Reviewer 2 Report

This manuscript discusses field studies to measure the prevalence of two important oyster pathogens, Perkinsus marinus and Haplosporidium nelson throughout the Atlantic coast of Georgia.  The authors investigate site variation as well as infection prevalence and intensity associations with condition factor and environmental parameters.  An update on the status of these two parasites in Georgia is important given that the last survey was almost 2 decades ago.  The manuscript is clearly written, with a logical flow and clear presentation of results.  However, several potential areas for improvement or further discussion were identified.  The majority of issues revolved around that statistical analyses, which were not appropriately justified given the types of data, making the results and conclusions questionable.  Specific comments and recommendations are as follows:

1)    It was surprising that the history of two parasites of study in Georgia, was not provided until the discussion section.  The study might be better motivated if this information was provided in the introduction.

2)    It was surprising that histology was not performed on oysters.  This would likely have provided a more accurate assessment of oyster condition index, parasite prevalence, and intensity, allowing for a more complete determination of the correlation between these factors

3)    Along these lines, PCR data should be interpreted carefully, given that it provides a quantification of DNA copies, which may not perfectly correlate with parasite loads.

4)    As the authors have indicated, measurement of environmental variables at one time point greatly limits the ability to infer environmental drivers in the data.  Many of the parameters measured can vary drastically throughout a given day.  As such the utility of this analysis is not clear.

5)    Along these lines, the use of continuous environmental parameter monitoring data on Sapelo island is a step in the right direction.  However, it is unclear why the average over 3 months was selected.  Likewise, why is an alpha of 0.1 justified because there are 4 sites?

6)     In general, the amount of site/creek replication is low.  Furthermore, it is unclear is all sites were sampled at the same time.  If not, temporal sampling could be affected some of the site differences observed.

7)    When analyzing prevalence data, logistic regression is considered a much better and more widely accepted approach than anova on arcsin sqrt transformed data.  This will also allow for the inclusion of random effects, such as creeks (discussed below).

8)    I’m concerned about the correlation matrix and the possibility of type I error inflation given the number of pairwise tests.  What do points in figure represent?  It appears they are mean values of reefs within creeks. There seems to be a lack of consideration of psuedoreplication within creeks.  A better approach would be to use generalized linear models include creek as a random effect in the models.

9)    It is unclear why the two condition index analyses can’t be combined, with two predictor factors: creek and presence absence parasite.

10) Was coinfection higher or lower than expected by chance.  Can be calculated easily by expected co-infection = prevalence single infection MSX * prevalence single infection dermo.

11) It would be interesting to see the individual oyster variation in condition index.  I expect extremely low condition oysters are rare, which may indicate mortality at low condition indexes.  This may be masking correlations between infection intensity and condition index.

12)  Larger oysters also likely to be more resistant because young susceptible oysters may die off.

Author Response

Reviewer 2

This manuscript discusses field studies to measure the prevalence of two important oyster pathogens, Perkinsus marinus and Haplosporidium nelson throughout the Atlantic coast of Georgia.  The authors investigate site variation as well as infection prevalence and intensity associations with condition factor and environmental parameters.  An update on the status of these two parasites in Georgia is important given that the last survey was almost 2 decades ago.  The manuscript is clearly written, with a logical flow and clear presentation of results.  However, several potential areas for improvement or further discussion were identified.  The majority of issues revolved around that statistical analyses, which were not appropriately justified given the types of data, making the results and conclusions questionable.  Specific comments and recommendations are as follows:

 We appreciate the constructive feedback provided by the Reviewer and completely reworked the statistical analysis used for our data. While the overall results or interpretations did not change, we feel using the analytical approaches described by the Reviewer now allows our arguments to be stronger.

1)    It was surprising that the history of two parasites of study in Georgia, was not provided until the discussion section.  The study might be better motivated if this information was provided in the introduction.

We regret this oversight in the initial version and have been clear about the history of these parasites in Georgia throughout the introduction.

2)    It was surprising that histology was not performed on oysters.  This would likely have provided a more accurate assessment of oyster condition index, parasite prevalence, and intensity, allowing for a more complete determination of the correlation between these factors

While we agree that the traditional approach of histology is useful particularly for showing disease impacts and oyster pathological responses, real-time PCR is significantly more accurate and sensitive than traditional histology.  Histology misses low level infections, parasites that are not in an infective state, as well as high parasite loads that do not result in pathogenic impacts.  As such, using a qPCR approach actually allows us to identify parasite prevalence and parasite loads that would be missed using histology.  Importantly, the protocols, primers and probes we used have been designed and compared with traditional histology (i.e. Gauthier et al. 2006, Wilbur et al. 2012) and commonly used for disease screening (i.e. Watts 2018,  Hanley et al. 2019).

3)    Along these lines, PCR data should be interpreted carefully, given that it provides a quantification of DNA copies, which may not perfectly correlate with parasite loads.

We agree that real-time PCR detects gene copy number and not specifically parasite number. Because the parasites are unicellular, DNA copies are a very good proxy for parasite load, which we were able to quantify because we created standard curves with known quantities of DNA fragments. Importantly, these values represent relative abundances among the oysters screened, and does not change the fact that high copy numbers are high intensity infections.

4)    As the authors have indicated, measurement of environmental variables at one time point greatly limits the ability to infer environmental drivers in the data.  Many of the parameters measured can vary drastically throughout a given day.  As such the utility of this analysis is not clear.

Based on the Reviewers suggestion, we have completely redone the analysis.  And while we agree – and we made clear in the original manuscript – that single time point limits our ability to make STRONG statements regarding environmental drivers, it does not preclude us from exploring potential drivers.  There are fairly consistent water quality differences between creeks – particularly in regards to salinity – and our previous work in some of these creeks highlight fairly consistent gradients within the survey creeks (i.e. Carroll et al. 2021) that are driven by geomorphic features, groundwater, etc.  So on the one hand, yes, some of these values, like DO, may change quite a bit over the course of the day, the physical and geological features of these creeks lead to areas where DO is consistently higher or lower relative to each other.  Importantly, we are not anywhere in this manuscript attempting to attain values to assign as good or bad in regards of parasite prevalence or intensity.  Rather, we are trying to ascertain whether there are any patterns or trends (i.e. relationships) between disease causing parasites and water quality.

5)    Along these lines, the use of continuous environmental parameter monitoring data on Sapelo island is a step in the right direction.  However, it is unclear why the average over 3 months was selected.  Likewise, why is an alpha of 0.1 justified because there are 4 sites?

We have redone the analysis to assuage any reviewer concerns.  Additionally, 3 months was selected because the parasites are most intense during the summer, and we took the WQ values for the entire summer season up to collection, and we have clarified this in the methods.

6)     In general, the amount of site/creek replication is low.  Furthermore, it is unclear is all sites were sampled at the same time.  If not, temporal sampling could be affected some of the site differences observed.

We described the sampling regime in the methods, but have now clarified.  In coastwide sampling, all oysters were collected within 2 weeks of each other.  In this sampling, these creeks were KMs long and the closest sampled reefs were at least 500 m apart from each other. On Sapelo, all the oysters were collected on the same day.

7)    When analyzing prevalence data, logistic regression is considered a much better and more widely accepted approach than anova on arcsin sqrt transformed data.  This will also allow for the inclusion of random effects, such as creeks (discussed below).

8)    I’m concerned about the correlation matrix and the possibility of type I error inflation given the number of pairwise tests.  What do points in figure represent?  It appears they are mean values of reefs within creeks. There seems to be a lack of consideration of psuedoreplication within creeks.  A better approach would be to use generalized linear models include creek as a random effect in the models.

In response to both point 7 and 8, prevalence data have been re-analyzed with a logistic mixed model to model the binomial presence/ absence data of parasite infection more accurately and to allow for the inclusion of random effects, such as creeks and sites within creeks. Additional linear mixed models were included to predict parasite intensity as well as oyster metrics (see point 9).

9)    It is unclear why the two condition index analyses can’t be combined, with two predictor factors: creek and presence absence parasite.

As noted above, for the coastwide data, linear mixed models were included for two oyster metrics: condition and length. Both models included parasite prevalence and creek as fixed effects and site as a random effect. The Sapelo creeks were analyzed with linear models due to the absence of random effects.

10) Was coinfection higher or lower than expected by chance.  Can be calculated easily by expected co-infection = prevalence single infection MSX * prevalence single infection dermo.

Coinfection was higher than expected, but only in the coastwide data set.  We hypothesize this is because the prevalence of P. marinus was overall much higher in the Sapelo data set (82%) than the coastwide data set (61%).  We have conducted chi-square tests and now incorporated those into the results.

11) It would be interesting to see the individual oyster variation in condition index.  I expect extremely low condition oysters are rare, which may indicate mortality at low condition indexes.  This may be masking correlations between infection intensity and condition index.

We have redone the analysis with each individual oyster within sites within creeks.  We have high and low condition oysters in the data set. Further, given environmental factors (i.e. salinity) had a large effect on oyster condition in our data set, it is improper to assume that any low condition oysters are from parasites, as our data suggests otherwise.

12)  Larger oysters also likely to be more resistant because young susceptible oysters may die off.

Literature suggests that for Dermo, larger and older oysters that are most affected because the mode of transmission is via the water column and they are processing more water.  When we reanalyzed the data using the Reviewer suggestions, we found that, specifically in our Sapelo Island data set, that larger oysters tended to have higher prevalence and intensity of P. marinus. There is no evidence to suggest that smaller oysters are more susceptible or are dying off.

Round 2

Reviewer 1 Report

None

Reviewer 2 Report

All comments have been adequately addressed.  It may be worth considering that condition index was not found to be significant because oysters will low condition succumb to mortality and are not observed.